# Cullin 1 (CUL1) Promotes Primary Ciliogenesis through the Induction of Ubiquitin-Proteasome-Dependent Dvl2 Degradation

**DOI:** 10.3390/ijms22147572

**Published:** 2021-07-15

**Authors:** Sun-Ok Kim, Kyoung Sang Cho, Bo Yeon Kim, Kyung Ho Lee

**Affiliations:** 1Anticancer Agent Research Center, Korea Research Institute of Bioscience and Biotechnology (KRIBB), 30 Yeongudanji-ro, Ochang, Cheongju 28116, Korea; sunok@kribb.re.kr; 2Department of Biological Sciences, Konkuk University, Seoul 05029, Korea; kscho@konkuk.ac.kr; 3Department of Biomolecular Science, University of Science and Technology, Daejeon 34113, Korea

**Keywords:** Dvl2, CUL1, primary ciliogenesis, ubiquitination, proteolysis

## Abstract

Primary cilia are nonmotile cellular signal-sensing antenna-like structures composed of microtubule-based structures that distinguish them from motile cilia in structure and function. Primary ciliogenesis is regulated by various cellular signals, such as Wnt, hedgehog (Hh), and platelet-derived growth factor (PDGF). The abnormal regulation of ciliogenesis is closely related to developing various human diseases, including ciliopathies and cancer. This study identified a novel primary ciliogenesis factor Cullin 1 (CUL1), a core component of Skp1-Cullin-F-box (SCF) E3 ubiquitin ligase complex, which regulates the proteolysis of dishevelled 2 (Dvl2) through the ubiquitin-proteasome system. Through immunoprecipitation-tandem mass spectrometry analysis, 176 Dvl2 interacting candidates were identified, of which CUL1 is a novel Dvl2 modulator that induces Dvl2 ubiquitination-dependent degradation. Neddylation-dependent CUL1 activity at the centrosomes was essential for centrosomal Dvl2 degradation and primary ciliogenesis. Therefore, this study provides a new mechanism of Dvl2 degradation by CUL1, which ultimately leads to primary ciliogenesis, and suggest a novel target for primary cilia-related human diseases.

## 1. Introduction

Primary cilia are nonmotile cilia and are structures that function as cellular antennae found on the cell surface [1,2]. It does not move but is known to transmit signals from outside the cell to the inside, regulating various cellular functions, such as cell development, growth, differentiation, and homeostasis [3]. Defects in the formation, maintenance, and function of primary cilia cause a number of diseases and developmental disorders known as ciliopathies and have recently been reported to be deeply related to the onset of cancer [3,4].

The primary cilia begin to be formed after the end of mitosis (G0 stage) and break down upon entering mitosis. Elaborate mechanisms that regulate the formation and reabsorption of primary cilia are being studied. The centrosome is an organelle that plays a very important role in this process. Therefore, a series of powerful regulatory mechanisms in the centrosome are required to control appropriate cilia assembly and disassembly in cells. Studies on centrosomes that perform these functions and cell signaling pathways that regulate cilia components are being conducted. In particular, the functions of Wnt signaling related to centrosomal proteins in primary cilia assembly and disassembly have been reported recently [2,5,6]. Wnt3a, a canonical Wnt ligand, stimulation generates β-catenin phosphorylation, resulting in the rearrangement of centriolar satellites that promotes primary ciliogenesis from the centrosome. In contrast to this process, stimulation of Wnt5a, a noncanonical Wnt ligand, causes centrosomal dishevelled 2 (Dvl2) phosphorylation, resulting in human enhancer filamentation 1 (HEF1) stabilization that induces primary cilia disassembly [2,5,6]. In addition, the critical role of protein degradation at the centrosome in primary cilia regulation has been reported. Some protein kinases, such as tau tubulin kinase 2 (TTBK2) and microtubule affinity regulating kinase 4 (MARK4), play a role in inducing cilia formation by removing CP110 from mother centriole [7,8,9]. Recently, autophagy-mediated proteolysis has been reported to promote cilia formation by removing oral-facial-digital 1 (OFD1) from centriolar satellites [10]. A subset of ubiquitin E3 ligase, including von Hippel-Lindau protein (pVHL) and mindbomb E3 ubiquitin protein ligase 1 (MIB1), has also been reported to regulate ciliary formation [11,12].

Cullin 1 (CUL1) is a component of the Skp1-Cullin-F-box (SCF) E3 ubiquitin ligase complex. SCF E3 ligases are the largest family of E3 ubiquitin ligases and include a substrate recognition subunit F-box protein, an adaptor protein Skp1, a RING protein RBX1/RBX2, and a scaffold protein CUL1. Binding of the F-box protein to the N-terminus of CUL1 is mediated by the adaptor protein Skp1 and is important for determining the substrate specificity of the SCF E3 ligase. On the other hand, binding of RBX1/RBX2 to the C-terminus of CUL1 constitutes ligase activity [13]. The Cullin-RING E3 ubiquitin ligase complex targets various substrates and affects various biological processes, including cell growth, development, signaling pathways, transcriptional regulation, genomic integrity, and tumor suppression [13]. The centrosomal function of CUL1 has been reported. For example, the regulation of polo-like kinase 4 (Plk4) levels through CUL1 activity, which inhibits the biogenesis of excessive daughter centrioles, exerts tumor suppressor function in cells [14]. However, to date, the primary cilia-related CUL1 functions have not yet been reported.

Dvl is a major component of the Wnt signaling pathway, which functions in both canonical and noncanonical Wnt signaling. The various Dvl functions related to Wnt signaling include cell migration, cell polarity, cell fate decision, cell proliferation, and primary cilia disassembly [5,15,16,17,18,19]. It has been reported that Dvl2 phosphorylation induced by Wnt5a stimulation triggers HEF1 stabilization, leading to primary cilia disassembly [5]. Casein kinase 1ε (CK1ε) activation by Wnt5a stimulation generates Dvl2 p-S143/p-T224 epitopes, phosphorylated Dvl2 makes a complex with polo-like kinase 1 (Plk1), and the Dvl2-Plk1 complex recruits HEF1 from the HEF1 destruction complex. Finally, these sequential events trigger HEF1 stabilization and HDAC6 activation, leading to primary cilia disassembly [5,20]. More than 50 phosphorylation sites are phosphorylated by a variety of kinases, such as NIMA-related kinase 2 (NEK2), CK1δ/ε, CK2, receptor interacting serine/threonine kinase 4 (RIPK4), PAR-1, and Abl in Dvl proteins [21,22,23,24,25,26,27]. The diverse phosphorylations of Dvl produce various states of Dvl2 proteins, which is involved in various cellular events. The cellular Dvl2 level is critically regulated by either the ubiquitin-proteasomal or autophagy-lysosomal pathway [28,29,30]. Various E3 ubiquitin ligases targeting Dvl2, such as kelch-like family member 12 (KLHL12)-CUL3, itchy E3 ubiquitin protein ligase (ITCH), NEDD4-like E3 ubiquitin protein ligase (NEDD4L), Malin, pVHL, and neuregulin receptor degradation protein 1 (Nrdp1), have been reported to negatively regulate the Wnt signaling pathway [31,32,33,34,35]. However, to date, CUL1-dependent Dvl2 regulation and its intracellular functions have not yet been identified.

This study has identified CUL1 as a novel Dvl2 modulator and CUL1-dependent Dvl2 degradation leading to primary ciliogenesis. These findings may provide a novel mechanistic insight into studies involving primary ciliogenesis.

## 2. Results

### 2.1. Identification of CUL1 as a New Dvl2 Binding Partner

To identify novel Dvl2 interacting proteins, Flag-Dvl2-overexpressing cells were immunoprecipitated by an anti-Flag antibody, and coimmunoprecipitated proteins were identified by liquid chromatography (LC)–tandem-mass spectrometry (MS/MS; Figure 1A). Excluding putative nonspecific binding noise proteins, such as various immunoglobulin chains, among 176 significant identified binding candidates, CUL1, a core component of SCF E3 ubiquitin ligase complexes, efficiently binds to Dvl2 (Appendix A). In addition, STRING analysis performed using the identified binding candidates revealed no reports of a direct functional association between Dvl2 and CUL1 (Figure 1A). Subsequent analyses of the interaction between endogenous Dvl2 and CUL1 revealed that these two endogenous proteins interact in a variety of cell lines (Figure 1B), and this interaction was also confirmed in a reciprocal immunoprecipitation (IP) assay (Figure 1C). The specific interaction between Dvl2 and CUL1 was also confirmed in a GST pull-down assay using bacterially purified GST-CUL1 or GST-Dvl2. Compared with the respective GST-only pulled-down control, Flag-Dvl2 or Flag-CUL1 was precipitated along with GST-CUL1 or GST-Dvl2, respectively (Figure 1D,E). These results led to the conclusion that CUL1 is a novel binding partner of Dvl2.

### 2.2. CUL1 Activity Downregulates Dvl2 through the Ubiquitin-Proteasome System

Because CUL1 is known as a component of E3 ubiquitin ligase, Dvl2 protein ubiquitination by CUL1 was monitored. Flag-CUL1 wild-type (WT) expression significantly increased the high molecular weight Myc-Dvl2 ubiquitin ladder in the in vivo ubiquitination assay compared with the vector expression control (Figure 2A). Conversely, endogenous CUL1 depletion reduces Dvl2 ubiquitination (Figure 2B). Three different *CUL1* small interfering RNA (siRNA) transfections resulted in a significant reduction in the high molecular weight Dvl2 ubiquitin ladder compared with control siRNA transfection in the in vivo ubiquitination assay. Based on these observations, the change in the amount of Dvl2 by CUL1 was monitored. To this end, three different amounts of *pFlag-CUL1 WT* plasmid were transfected into HEK293T cells, and the change in the amount of Myc-Dvl2 was observed accordingly. There was an inverse correlation between the amount of CUL1 and Dvl2. As transfected *pFlag-CUL1 WT* increased, the amount of Myc-Dvl2 decreased (Figure 2C). The 1 μg transfection of *pFlag-CUL1 WT* led to less than half the amount of Myc-Dvl2 compared with empty vector (Figure 2C, “0”) transfection. In addition, the 2 μg transfection of *pFlag-Dvl2 WT* almost eliminated Myc-Dvl2 (Figure 2C). Conversely, endogenous CUL1 depletion using *CUL1* siRNA (siCUL1) transfection caused an increase in endogenous Dvl2. CUL1 depletion resulted in an approximately two-fold increase in endogenous Dvl2 compared with control siRNA (siCont) transfection (Figure 2D). In addition, a lower percentage of sodium dodecyl sulfate-polyacrylamide gel electrophoresis (SDS-PAGE; 8%) revealed the interesting phenomenon that hyperphosphorylated Dvl2 (Figure 2D, “b”) [5,36] was significantly increased by CUL1 depletion. The phosphorylation status of endogenous Dvl2 was divided into two different forms, one is fast-migrating (nonphosphorylated/less phosphorylated; referred to as the “a-form”) and the other is slow-migrating (hyperphosphorylated; referred to as the “b-form”) [5]. This may reflect the possibility that the main target of CUL1-dependent Dvl2 degradation is not the nonphosphorylated/less phosphorylated form of Dvl2 but rather the hyperphosphorylated form of Dvl2. To confirm this possibility, hereafter, endogenous Dvl2 was detected using both high (12%) and low (8%) SDS-PAGE in subsequent experiments.

Because CUL1 is known as a component of E3 ubiquitin ligase [37], and the cellular level of Dvl2 is regulated by the ubiquitin-proteasomal and autophagy-lysosomal pathway [28,29,30], Dvl2 regulation by CUL1 may be involved in the ubiquitin-proteasome pathway. To verify this hypothesis, various proteolysis pathway inhibitors were tested. Interestingly, although proteasome inhibitors (MG132, ALLN, and lactacystin) and autophagy inhibitors (bafilomycin A1 and chloroquine) stabilized the total amount of Dvl2, which is shown in 12% SDS-PAGE, compared with dimethyl sulfoxide (DMSO) treatment, hyperphosphorylated Dvl2 (Figure 2E, “b” shown in 8% SDS-PAGE) was stabilized by only proteasome inhibitors. In contrast, nonphosphorylated/less phosphorylated Dvl2 (Figure 2E, “a”) was stabilized by only autophagy inhibitors (Figure 2E). From this result and Figure 2D, it was speculated that hyperphosphorylated Dvl2 is a proteasome-dependent target and nonphosphorylated/less phosphorylated Dvl2 is an autophagy-dependent target in Dvl2 proteolysis. CUL1-induced Dvl2 ubiquitination was dramatically increased by proteasome inhibition but not by autophagy inhibition (Figure 2F). Treatment with MG132, ALLN, or lactacystin significantly accumulated a high molecular weight Myc-Dvl2 ubiquitin ladder by Flag-CUL1 expression in the in vivo ubiquitination assay compared with the DMSO control. However, treatment with bafilomycin A1 or chloroquine did not efficiently accumulate the Dvl2 ubiquitin ladder. These results supported the idea that CUL1-dependent Dvl2 ubiquitination is involved in the ubiquitin-proteasome pathway.

### 2.3. CUL1 Neddylation Is Essential for CUL1-Dependent Dvl2 Ubiquitination

The covalent attachment of NEDD8 proteins to the C-terminus of CUL1, called neddylation, plays a key role in CUL1-based SCF E3 ubiquitin ligase activity [38,39]. Therefore, in subsequent analysis, to clarify CUL1 activity-dependent Dvl2 ubiquitination, a neddylation-deficient CUL1 deletion mutant (CUL1 DM) similar to the dominant-negative form of CUL1 [40] was generated (Figure 3A), and the pharmaceutical inhibition of neddylation by MLN4924 [41] treatment was performed. Dvl2 ubiquitination induced by either CUL1 WT or CUL1 DM was monitored. Flag-CUL1 WT expression significantly increased Myc-Dvl2 ubiquitination in HEK293T cells, whereas Flag-CUL1 DM expression did not increase the Dvl2 ubiquitin ladder compared with empty vector (Figure 3B, “-”) transfection (Figure 3B). The Dvl2 ubiquitination level by Flag-CUL1 WT expression was significantly higher than Flag-CUL1 DM or vector control expression.

Next, CUL1 activity-dependent Dvl2 degradation was confirmed by comparing CUL1 WT and CUL1 DM mutant expression. To this end, different amounts of *pFlag-CUL1 WT* or *DM* plasmids were co-transfected with *pMyc-Dvl2* into HEK293T cells. Myc-Dvl2 levels gradually decreased with a gradual increase in Flag-CUL1 WT expression. However, a gradual increase in the Flag-CUL1 DM mutant failed to significantly alter Myc-Dvl2 levels (Figure 3C). In addition, the pharmaceutical inhibition of neddylation by MLN4924 treatment restored Dvl2 levels and reduced Dvl2 ubiquitination. MLN4924 is known as a NEDD8-activating enzyme (NAE) inhibitor [42]. To inhibit CUL1 neddylation, which is essential for CUL1 activity, cells were treated with varying amounts of MLN4924. As a consequence of the compound treatment, the endogenous Dvl2 level was increased by MLN4924 treatment compared with the control DMSO treatment (Figure 3D, “0”). In addition, interestingly, hyperphosphorylated Dvl2 (Figure 3D, “b”) was increased by MLN4924 treatment. Thus, as speculated in Figure 2D,E, this may also reflect the possibility that hyperphosphorylated Dvl2 is a major target of CUL1-dependent Dvl2 ubiquitinylation. Consistent with this observation, Myc-Dvl2 ubiquitination also gradually decreased depending on the concentration of MLN4924 treatment in the in vivo ubiquitination assay (Figure 3E). HEK293T cells transfected with *pFlag-CUL1 WT*, *pMyc-Dvl2*, and *pHA-Ub* were treated with different amounts of MLN4924, after which Myc-Dvl2 ubiquitination was detected through the following in vivo ubiquitination assay. Dvl2 ubiquitination ladders were decreased by MLN4924 treatment compared with DMSO control (Figure 3E). 

### 2.4. Centrosomal Dvl2 Is Regulated by CUL1

To investigate the functional relationship between Dvl2 and CUL1, subcellular localizations of the two proteins were monitored. Both proteins were observed throughout the cytoplasm, and the extraction with detergent (Triton X-100; see Materials and Methods for details) before fixation clearly showed the centrosomal colocalization of the two proteins (Figure 4A). Subsequent analyses showed that CUL1 depletion significantly increased the centrosomal Dvl2 level. hTERT-RPE cells transfected with *CUL1* siRNA (siCUL1) showed a significant increase in endogenous Dvl2 signals at the centrosome compared with control siRNA (siCont.)-transfected cells (Figure 4B,C). In contrast, Flag-CUL1 WT expression significantly diminished the centrosomal Dvl2 level. hTERT-RPE cells transfected with *pFlag-CUL1 WT* showed a significant reduction in the signal intensity of centrosomal Dvl2 compared with empty vector or *pFlag-CUL1 DM* mutant transfected cells (Figure 4D,E). These data suggested that the function of centrosomal Dvl2 is likely to be regulated by CUL1 activity.

### 2.5. CUL1 Promotes Primary Ciliogenesis through Dvl2 Downregulation

It has been previously suggested that centrosomal Dvl2 mediates primary cilia disassembly [5]. Therefore, whether centrosomal CUL1 regulates primary ciliogenesis was investigated. To this end, primary ciliogenesis under CUL1 depletion conditions was first monitored. hTERT-RPE cells transfected with either control siRNA (siCont.) or *CUL1* siRNA (siCUL1) were serum-starved to generate primary cilia according to the experimental schedule in Figure 5A (left). As expected, CUL1 depletion resulted in Dvl2 stabilization and, in particular, increased hyperphosphorylated Dvl2 as observed in Figure 2 and Figure 3 (Figure 5A, right, “b”). Moreover, a significantly reduced primary ciliogenesis was observed in *CUL1* knockdown cells (siCUL1) compared with control cells (siCont.; Figure 5B,C). Control siRNA transfection resulted in more than 50% primary cilia containing cells in this experimental condition; however, *CUL1* siRNA transfection caused less than 25% of cells with primary cilia (Figure 5B,C). In addition, the average length of primary cilia in *CUL1* knockdown cells was shorter than that of control knockdown cells (Figure 5D,E). The average length of primary cilia in control or *CUL1* siRNA-transfected cells was ~3.7 or 2.1 μm, respectively (Figure 5E).

In contrast to these observations, Flag-CUL1 WT overexpression downregulates endogenous Dvl2 (Figure 5F, bottom). Interestingly, unlike previous observations, both “a” and “b” forms of Dvl2 were affected by Flag-CUL1 overexpression. This is probably due to the excessive response resulting from the overexpression system. Flag-CUL1 WT expression leading to Dvl2 downregulation promoted primary ciliogenesis compared with empty vector or Flag-CUL1 DM mutant (Figure 5F–H). Under this experimental condition (Figure 5F, top), Flag-CUL1 WT expression resulted in more than 40% cells with primary cilia; however, the expression of either empty vector or Flag-CUL1 DM caused ~30% or 27% cells with primary cilia, respectively (Figure 5G,H). Additionally, Flag-CUL1 WT expression generated longer cilia (~3.4 μm long) than that of either empty vector (2.4 μm long) or Flag-CUL1 DM mutants (2.5 μm long; Figure 5I,J).

CUL1-dependent primary ciliogenesis was further confirmed by rescue experiments using Flag-CUL1 WT expression in endogenous CUL1 knockdown cells. Endogenous CUL1 was depleted in hTERT-RPE cells using *CUL1* siRNA transfection, and an empty vector, Flag-CUL1 WT, or Flag-CUL1 DM was expressed in *CUL1*-depleted cells. The resulting cells were serum-starved based on the experimental schedule described in Figure 6A (left). The increase in Dvl2 levels by endogenous *CUL1* knockdown was reduced by Flag-CUL1 WT (WT) expression, but the expression of vector control (-) or Flag-CUL1 DM (DM) did not decrease the Dvl2 levels (Figure 6A, right). Overexpression of Flag-CUL1 WT or DM altered both the “a” and “b” form of Dvl2 similar to that observed in Figure 5F. Consistent with Figure 5, endogenous *CUL1* depletion generated less ciliated cell population and shorter cilia than control knockdown cells (Figure 6B–E). Under these conditions, Flag-CUL1 WT expression restored both primary ciliated cell population and ciliary length. In *CUL1*-depleted cells, the less ciliated cell population (13%) and shortened primary cilia (2.4 μm) were recovered by CUL1 WT expression to 38% primary ciliated cell population and 4.5-μm long primary cilia (Figure 6C,E). However, the expression of either the empty vector or the Flag-CUL DM in *CUL1*-depleted cells did not efficiently restore the primary ciliary phenotype in both ciliated cell population and ciliary length (Figure 6B–E). Collectively, these data supported the idea that CUL1 promotes primary ciliogenesis through the ubiquitin-proteasome-dependent Dvl2 degradation induced by CUL1 activity.

## 3. Discussion

This study identified a novel function of CUL1, which ubiquitinylates Dvl2 and finally promotes primary ciliogenesis. CUL1 was isolated from LC-MS/MS analysis as a novel binding partner of Dvl2. CUL1 is a component of the SCF E3 ligase complex and plays an important role in protein ubiquitination-dependent degradation in various biological processes, including cell cycle, cell growth, development, signal transduction, and transcriptional control [13,37]. However, neither CUL1 function in primary ciliogenesis nor CUL1-dependent Dvl2 degradation has been reported.

In a previous study, the Wnt5a-CK1ε-Dvl2-Plk1-mediated primary cilia disassembly pathway was identified [5]. Although Dvl2 degradation was not reported in a previous study, CUL1-dependent Dvl2 degradation may antagonize the function of the Wnt5a-dependent primary cilia disassembly pathway through the induction of Dvl2 degradation. In addition, Wnt3a-dependent primary ciliogenesis was recently reported, but Dvl2 proteolysis was not considered in that study either [6]. The findings of this study may enhance Wnt3a-dependent primary ciliogenesis through the elimination of Dvl2 by CUL1 activity. It has been reported that the half-life of Dvl2 protein is regulated by ubiquitylation-dependent proteasomal degradation [32,33,34,35] as well as autophagy-lysosome-dependent degradation [29,43]. Both Dvl2 degradation machineries have been reported to be related to the Wnt signaling pathway [30]. Therefore, studies on Wnt3a/Wnt5a-dependent CUL1 activity regulation are worth conducting in the future, and it may give a functional linkage between this study and Wnt3a-dependent primary ciliogenesis/Wnt5a-dependent primary cilia disassembly.

Two different functions have been reported for cullin 3 (CUL3), another family member of cullin E3 ligase (31% amino acid identity with CUL1). One is CUL3-dependent Dvl degradation in the Wnt β-catenin signaling pathway [31], and the other is CUL3-mediated primary cilium formation through CEP97 degradation [44]. Therefore, although CUL3 was not identified as a Dvl2 binding candidate in our IP-LC-MS/MS experiments, and only 31% amino acid identity was found between CUL1 and CUL3, it is presumed that the two proteins may share similar cellular functions or play a compensating role in primary ciliogenesis. In addition, it cannot rule out the possibility that these findings may be incorporated into either Cullin3-KCTD10-CEP97-mediated primary cilia formation or KLHL12-Cullin3-Dvl2-mediated Wnt β-catenin pathway regulation. However, it is unknown whether CUL1-dependent machinery and CUL3-dependent machinery work together or independently, so further studies should investigate whether CUL3 and CEP97 are included in the CUL1-dependent Dvl2 degradation machinery found in this study.

The length of primary cilia and the number of cells with primary cilia were observed to be regulated by CUL1 in this study. Therefore, it would be interesting to study whether CUL1-dependent Dvl2 regulation acts on the initiation of primary ciliogenesis or the elongation of the primary cilium. As centrosomal CUL1 function on centriole duplication through Plk4 regulation and centrosomal Dvl2 function on centrosomal separation through Nek2 kinase regulation have been reported previously [14,22], it is possible that centrosomal CUL1 and Dvl2 work together not only in primary ciliogenesis but also in centriole duplication/centrosome separation during cell cycle progression. Therefore, it is possible that the functional connection between CUL1 and Dvl2 may exert its different functions in different cell cycle stages, such as primary ciliogenesis in G0, centriole duplication in G1/S, and centrosomal separation in the mitotic phase of the cell cycle.

Interestingly, this study found CUL1 activity-dependent regulation of the slow-migrating form of Dvl2 (hyperphosphorylated Dvl2; Figure 2D,E, Figure 3D, and Figure 5A, “b”). Depletion of endogenous CUL1 (Figure 2D and Figure 5A), proteasome inhibitor treatment (Figure 2E), and neddylation inhibitor treatment (Figure 3D) support the idea that the hyperphosphorylated form of Dvl2 is a target for CUL1-dependent proteasomal degradation. In contrast, autophagy inhibitor treatment regulated the fast-migrating form of Dvl2 (nonphosphorylated/less phosphorylated Dvl2; Figure 2E, “a”). Thus, nonphosphorylated/less phosphorylated Dvl2 may be a target for lysosomal degradation machinery. In our previous report [5,6], CK1δ/ε was responsible for the generation of slow-migrating form of Dvl2, thus it is worthwhile to investigate whether the activity of CK1δ/ε affects CUL1-dependent Dvl2 degradation through regulation of phosphodegrons recognized by SCF ligases. However, the overexpression of Flag-CUL1 WT or DM altered the both fast-migrating (“a”) and slow-migration (“b”) form of Dvl2 (Figure 5F and Figure 6A). We speculate that an excessive response by overexpression of CUL1 WT/DM could alter both patterns of Dvl2, or that starvation in the experimental process could affect the relevant proteolytic pathway, resulting in degradation of both Dvl2 forms. However, more detailed studies are needed in the future to precisely elucidate the major target form of Dvl2 for CUL1-dependent proteasomal degradation. 

These data suggested CUL1 as a new primary ciliogenesis factor regulating Dvl2 half-life and may incorporate into the Wnt-dependent primary ciliogenesis pathway. Thus, the findings may provide new basic knowledge in studying primary ciliary dynamics and cilia-related diseases.

## 4. Materials & Methods

### 4.1. Plasmid Construction

Human *CUL1* was amplified by polymerase chain reaction (PCR) from the reverse transcription (RT)-PCR product of HeLa CCL2 cells. Full-length of CUL1 cDNA was cloned into *Not*I-*Bam*HI site of *pFLAG-CMV-2* (Sigma, St. Louis, MO, USA) or *Bam*HI-*Not*I site of *pGEX-4T-2* vector (Amersham Biosciences, Piscataway, NJ, USA). Deletion mutant (DM, a.a 1-435) form of CUL1 was generated by PCR using *pFlag-CMV2-CUL1* plasmid as a template and sub-cloned into *Not*I-*Bam*HI site of *pFLAG-CMV-2* (Sigma, St. Louis, MO, USA) or *Bam*HI-*Not*I site of *pGEX-4T-2* (Amersham Biosciences, Piscataway, NJ, USA) vector. A *BamHI-EcoRI* fragment of mouse Dvl2 was generated by PCR and sub-cloned into *pFLAG-CMV-2*, *pGEX-4T-2*, or *pCMV-Tag 3B* vector, as described previously [5].

### 4.2. Cell Culture and Transfection

HEK293T, HeLa CCL2, and hTERT-RPE cells were cultured and maintained according to the recommendations of the American Type Culture Collection (ATCC, Manassas, VA, USA). 

Plasmids were transfected to corresponding cells by using X-tremeGENE HP DNA transfection reagent (Roche, Mannheim, Germany), and siRNAs were transfected to corresponding cells by using Lipofectamine RNAiMAX reagent (Invitrogen, Waltham, MA, USA) in accordance with the manufacturer’s instructions. The siRNA sequences targeting *CUL1* are: #1 5′-GACGAAGGACGAAAAGGAATT-3′, #2 5′-CAUUUUGGCGCAAGUUUUATT-3′, and #3 5′-CUAAACUUCAGCGCAUGUUTT -3′. Mixed *CUL1* siRNA was purchased from Santa Cruz Biotechnology (Santa Cruz, CA, USA). The *CUL1* siRNA #1 contains 5′-**GACGA**A**GG**A**CG**A**AAA**GGA**A**TT-3′ (from nt 448 to 466 in human *CUL1*) that bears thirteen different nucleotide sequences (bold and underlined) between *CUL1* and *CUL3*. The *CUL1* siRNA #2 contains 5′-CA**UU**UU**G**G**C**G**C**AA**G**UU**UU**ATT-3′ (from nt 1902 to 1920 in human *CUL1*) that bears eight different nucleotide sequences (bold and underlined) between *CUL1* and *CUL3*. The *CUL1* siRNA #3 contains 5′-C**U**AAACU**UC**A**GC**G**C**AUGUUTT-3′ (from nt 1505 to 1523 in human *CUL1*) that bears six different nucleotide sequences (bold and underlined) between *CUL1* and *CUL3*. After transfection, cells were subjected to various biochemical assays such as GST pull-down assay, immunoprecipitation, immunoblotting, or immunostaining analysis. Detailed information on siRNA was provided in Appendix A.

### 4.3. GST Pull-Down Assay

GST pull-down assay was performed as described previously [45]. Briefly, bead-bound GST-CUL1 WT or GST-Dvl2 was purified from bacteria (BL21) and incubated with HEK293T cell lysates expressing Flag-Dvl2 or Flag-CUL1 for 5 h, and then beads were precipitated by brief centrifugation. The precipitates were washed with 1 × TBSN buffer (20 mM Tris-Cl (pH 8.0), 150 mM NaCl, 1.5 mM EDTA, 0.5% NP-40, 5 mM EGTA, 10 mg/mL p-nitrophenyl phosphate (pNPP; Sigma-Aldrich, St. Louis, MO, USA), and protease inhibitor cocktail (Roche, Mannheim, Germany)) four times and subjected to immunoblotting analysis with indicated antibodies. All the antibodies used for immunoblotting analyses were summarized in Appendix A.

### 4.4. In Vivo Ubiquitination Assay

HEK293T cells were co-transfected with *pHA-ubiquitin* (Ub) (a gift from Kyung S. Lee, NIH, Bethesda, MD, USA), *pMyc-Dvl2*, and *pFlag-CUL1* construct. To prevent proteasomal degradation of Dvl2, cells were treated with 10 μM of MG132 for 3 h before harvest. Twenty-four hours after transfection, cells were harvested and used for immunoprecipitation with anti-Flag or anti-Myc antibody.

### 4.5. Immunoprecipitation and Immunoblotting Analysis

Immunoprecipitation and immunoblotting analysis were carried out as described previously [6]. For immunoprecipitation, HEK293T, hTERT-RPE, or HeLa CCL2 cells harvested with 1 × ice-cold phosphate-buffered saline (PBS) were lysed in 1 × TBSN buffer (20 mM Tris-Cl (pH 8.0), 0.5% NP-40, 150 mM NaCl, 1.5 mM EDTA, 5 mM EGTA, 10 mg/mL pNPP (Sigma-Aldrich, St. Louis, MO, USA), and a protease inhibitor cocktail (Roche, Mannheim, Germany)). The total cell lysates were then centrifuged at 15,000 rpm for 20 min at 4 °C. The cell lysates were incubated with the indicated antibodies for 4 h at 4 °C, and then protein A or G–Sepharose beads (Santa Cruz Biotechnology, Santa Cruz, CA, USA) were added and incubated for an additional 2 h at 4 °C. Beads were then precipitated by brief centrifugation and washed with 1 × TBSN four times; 2 × Laemmli sample buffer (120 mM Tris-Cl (pH 6.8), 4% SDS, 20% Glycerol, 10% 2-mercaptoethanol, and 0.02% bromophenol blue) was then added to the precipitated beads, and the beads were boiled for 15 min at 95~100 °C. 

For immunoblotting analysis, samples were separated by 8~12% sodium dodecyl sulfate-polyacrylamide gel electrophoresis (SDS-PAGE) and transferred to PVDF membrane (Amersham Biosciences, Piscataway, NJ, USA). Membranes were incubated with primary antibodies for 4 h at room temperature or overnight at 4 °C. Membranes were then washed extensively with 1 × TBST ((50 mM Tris-Cl (pH 7.4), 137 mM NaCl, 0.05% Tween 20)) more than three times. Membranes were incubated with horseradish peroxidase (HRP)-conjugated secondary antibodies (Cell signaling technology, Danvers, MA, USA) for 1 h at room temperature. After extensive washing with 1 × TBST, immunoreactive signals on the membrane were detected with an enhanced chemiluminescence (ECL) detection system (Pierce, Rockford, IL, USA). Information on the antibodies used in this study is provided in Appendix A.

### 4.6. Immunofluorescence Analysis

Immunofluorescence analysis and the measurement of primary cilia were carried out as described previously [6]. The hTERT-RPE cells were seeded on coverslip in 12-well plates. The cells were fixed with 4% paraformaldehyde after transfection and serum starvation according to the experimental schedule outlined in the main figure. Cells were then permeabilized in cold methanol. To detect centrosomal localization of CUL1 and Dvl2, hTERT-RPE cells were extracted with PEMT buffer (100 mM PIPES (pH 6.8), 5 mM EGTA, 1 mM MgCl_2_, 0.2% Triton X-100) for 30 sec prior to fixation. After fixation and permeabilization, samples were extensively washed with PBS containing 0.1% Triton X-100 (1 × PBST) four times and then incubated with primary antibodies for overnight at 4 °C. After a wash with 1 × PBST, samples were incubated with Texas Red/Alexa Fluor 488-conjugated secondary antibodies (Invitrogen, Carlsbad, CA, USA) for 1 h at room temperature; 1 μg/mL of 4′, 6′-595 diamidino-2-phenylindole (DAPI) (Sigma, St. Louis, MO, USA) solution was used to stain the DNA. The resulting coverslips were mounted on the glass slides with a Fluoro-Gel mounting medium (EMS, Hatfield, PA, USA). 

Fluorescent images were observed and photographed using Zeiss AxioObserver (Carl Zeiss, Göttingen, Germany) fluorescence microscope system. To measure the length of primary cilium, images were acquired by Zeiss AxioObserver Z1 microscope (Carl Zeiss, Göttingen, Germany) at 1388 × 1040 pixels and 12-bit resolution and analyzed by ZEN v2.1 (Carl Zeiss, Göttingen, Germany) software. Antibodies used for immunofluorescence analysis are listed in Appendix A.

### 4.7. Inhibitor Treatment

To inhibit proteasomal degradation pathway, cells were treated with 10 μM of MG132 (Calbiochem, San Diego, CA, USA), 100 μM of ALLN (Sigma, St. Louis, MO, USA), or 10 μM of lactacystin (Sigma, St. Louis, MO, USA) for 3 h. To inhibit autophagy, cells were treated with 0.2 μM of bafilomycin A1 (Sigma, St. Louis, MO, USA) or 100 μM of chloroquine (Sigma, St. Louis, MO, USA) for 3 h. To inhibit NEDD8-activating Enzyme (NAE), cells were treated with 0.5 or 1 μM of MLN4924 (Calbiochem, San Diego, CA, USA). Detailed information on chemicals is provided in Appendix A.

### 4.8. In-Gel Digestion with Trypsin and Extraction of Peptides

Protein bands from SDS-PAGE gels were excised and in-gel digested with trypsin according to established procedures [46] (ProteomeTech, Inc., Seoul, Korea). In brief, protein bands were excised from stained gels and cut into pieces. The gel pieces were washed for 1 h at room temperature in 25 mM ammonium bicarbonate buffer, pH 7.8, containing 50% (*v/v*) acetonitrile (ACN; Sigma-Aldrich, St. Louis, MO, USA). Following the dehydration of gel pieces in a centrifugal vacuum concentrator (Biotron, Inc., Incheon, Korea) for 10 min, gel pieces were rehydrated in 50 ng of sequencing grade trypsin solution (Promega, Madison, WI, USA). After incubation in 25 mM ammonium bicarbonate buffer, pH 7.8, at 37 °C overnight, the tryptic peptides were extracted with 100 μL of 1% formic acid (FA) containing 50% (*v/v*) ACN for 20 min with mild sonication. The extracted solution was concentrated using a centrifugal vacuum concentrator. Prior to mass spectrometric analysis, the peptides solution was subjected to a desalting process using a reversed-phase column [47]. In brief, after an equilibration step with 10 μL of 5% (*v/v*) formic acid, the peptides solution was loaded on the column and washed with 10 μL of 5% (*v/v*) formic acid. The bound peptides were eluted with 8 μL of 70% ACN with 5% (*v/v*) formic acid.

### 4.9. Identification of Proteins by LC-MS/MS

LC–MS/MS analysis was performed through nano ACQUITY UPLC and LTQ-orbitrap-mass spectrometer (Thermo Electron, San Jose, CA, USA) (ProteomeTech, Inc., Seoul, Korea). The column used BEH C18 1.7 μm, 100 μm × 100 mm column (Waters, Milford, MA, USA). The mobile phase A for the LC separation was 0.1% formic acid in deionized water and the mobile phase B was 0.1% formic acid in acetonitrile. The chromatography gradient was set up to give a linear increase from 10% B to 40% B for 21 min, from 40% B to 95% B for 7 min, and from 90% B to 10% B for 10 min. The flow rate was 0.5 μL/min. For tandem mass spectrometry, mass spectra were acquired using data-dependent acquisition with full mass scan (300–2000 m/z) followed by MS/MS scans. Each MS/MS scan acquired was an average of one microscans on the LTQ. The temperature of the ion transfer tube was controlled at 275 °C and the spray was 2.0 kV. The normalized collision energy was set at 35% for MS/MS. The individual spectra from MS/MS were processed using the SEQUEST software (Thermo Quest, San Jose, CA, USA) and the generated peak lists were used to query in house database using the MASCOT program (Matrix Science Ltd., London, UK). We set the modifications of Carbamidomethyl (C), Deamidated (NQ), Oxidation (M) for MS analysis and tolerance of peptide mass was 10 ppm. MS/MS ion mass tolerance was 0.8 Da, allowance of missed cleavage was 2, and charge states (+2, +3) were taken into account for data analysis. We took only significant hits as defined by MASCOT probability analysis.

### 4.10. Statistical Analysis

Statistical analyses were performed using Excel (Microsoft, Redmond, WA, USA) for unpaired two-tailed *t*-test of Figure 4C and Figure 5C,E, and SigmaPlot 13.0 (Systat Software, San Jose, CA, USA) for one-way ANOVA of Figure 4E, Figure 5H,J and Figure 6C,E.

## Figures and Tables

**Figure 1 ijms-22-07572-f001:**
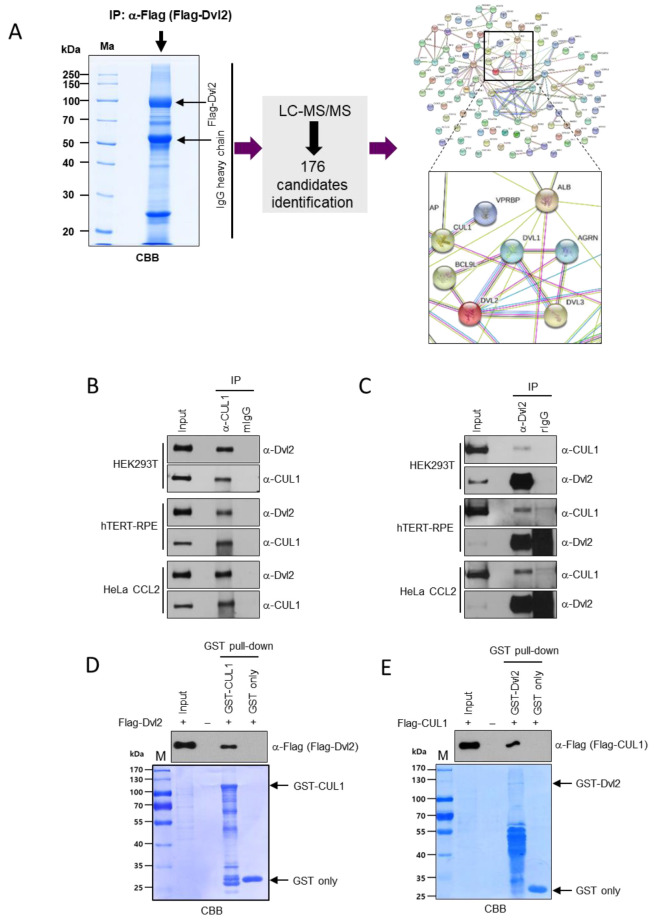
Dvl2 interacts with CUL1 E3 ligase. (**A**) HeLa cells transfected with *pFlag-tagged Dvl2 WT* were harvested and subjected to an IP assay with an anti-Flag antibody. The precipitates were separated using 10% SDS-PAGE, and stained bands were excised from the gel (**left**) and subsequently analyzed by MS to identify the interacting proteins (**middle**). The STRING protein-protein interaction network represents the interactions between 176 proteins isolated from LC-MS/MS (**right**). Different colors represent different kinds of evidence of the connection between proteins (**right**). In the STRING network, each line color represents the following meaning: green, neighborhood; red, gene fusion; blue, co-occurrence; black, coexpression; pink, experimental evidence; turquoise, database evidence; light green, evidence from text mining; violet, homology between the two proteins. Note that there is no direct connection between CUL1 and Dvl2 in the STRING network. (**B**,**C**) Physical interaction between endogenous CUL1 and Dvl2 in asynchronously growing HEK293T, hTERT-RPE, or HeLa CCL2 cell lines. Total cell lysates from various cells were subjected to IP with anti-CUL1 (**B**) or anti-Dvl2 (**C**) antibodies. The resulting immunoprecipitates were separated using 4–20% SDS-PAGE and immunoblotted with the indicated antibodies. Note that CUL1 and Dvl2 form a complex at the endogenous level in various cell lines. (**D**) HEK293T cells transfected with *pFlag-Dvl2* were subjected to a GST pull-down assay with either bacterially purified GST or GST-tagged CUL1. Pulled-down Flag-Dvl2 with GST-CUL1 was confirmed by immunoblotting with an anti-Flag antibody. CBB represents the amount of proteins. (**E**) HEK293T cells transfected with *pFlag-CUL1* were subjected to a GST pull-down assay with either bacterially purified GST or GST-tagged Dvl2. Pulled-down Flag-CUL1 with GST-Dvl2 was confirmed by immunoblotting with an anti-Flag antibody. CBB represents the amount of proteins. All data show representative images of three independent experiments.

**Figure 2 ijms-22-07572-f002:**
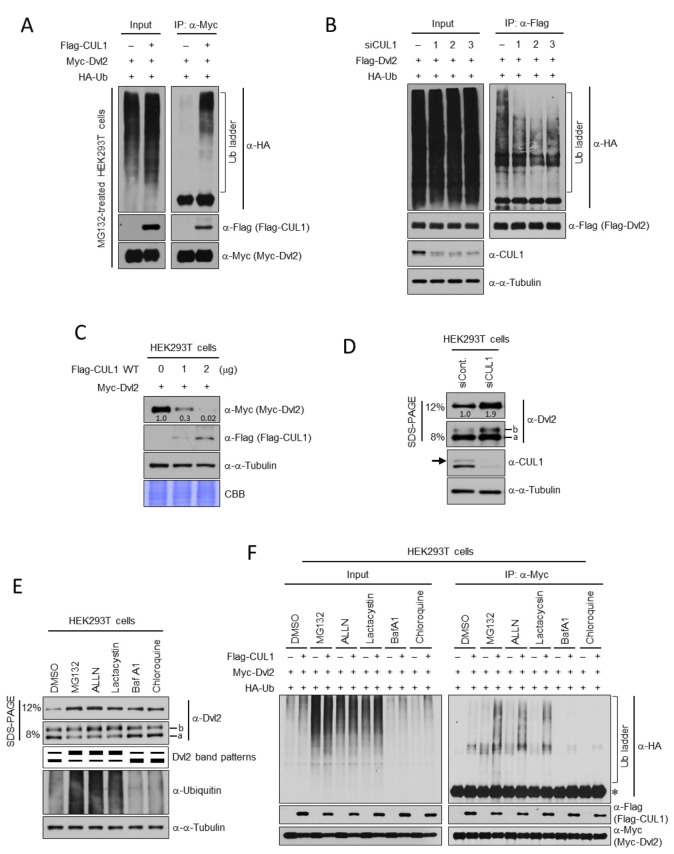
CUL1 mediates proteasome-dependent Dvl2 degradation. (**A**) HEK293T cells cotransfected with *pFlag-CUL1 WT*, *pMyc-Dvl2*, and *pHA-ubiquitin* (Ub) were treated with 10 μM MG132 for 3 h before cell harvest. Total cell lysates were subjected to IP with an anti-Myc antibody. The resulting immunoprecipitates were separated using 6% SDS-PAGE and immunoblotted with the indicated antibodies. (**B**) HEK293T cells were transfected with either control siRNA (-) or *CUL1* siRNA (three different siRNA sequences referred to as 1, 2, or 3). After 24 h siRNA transfection, cells were then cotransfected with *pFlag-Dvl2* and *pHA-Ub*. The resulting cell lysates were subjected to IP with an anti-Flag-antibody, and the immunoprecipitated beads were separated in 6% SDS-PAGE followed by immunoblotting analysis with the indicated antibodies. (**C**) HEK293T cells cotransfected with *pFlag-CUL1 WT* and *pMyc-Dvl2* were subjected to immunoblotting analysis with the indicated antibodies. To monitor the effect of CUL1 on Dvl2 degradation, three different amounts of *pFlag-CUL1 WT* plasmids (0, 1, or 2 μg) were transfected to cells; 0 μg represents empty vector transfection. The band intensities of Myc-Dvl2 were measured using ImageJ and normalized to α-tubulin, and the relative values are marked below the bands. A constant amount of total DNA was transfected into the cells. (**D**) Total cell lysates from HEK293T cells transfected with either control (siCont.) or *CUL1* siRNA (siCUL1) for 24 h were subjected to immunoblotting analysis with the indicated antibodies. Immunoblotting with the anti-α-tubulin antibody represents the loading control of proteins. The band intensities of endogenous Dvl2 were measured using ImageJ and normalized to α-tubulin, and the relative values are marked below the bands. The arrow indicates neddylated CUL1. Fast- and slow-migrating Dvl2 forms are referred to as the “a” and “b” forms, respectively. (**E**) Asynchronously growing HEK293T cells were treated with the indicated compounds for 3 h to inhibit proteasome (MG132, ALLN, or lactacystin) or autophagy (bafilomycin A1 or chloroquine). The resulting cell lysates were subjected to immunoblot analysis with the indicated antibodies. Fast- and slow-migrating Dvl2 forms are referred to as the “a” and “b” forms, respectively. (**F**) Asynchronously growing HEK293T cells cotransfected with *pFlag-CUL1 WT*, *pMyc-Dvl2*, and *pHA-Ub* were treated with either proteasome inhibitor (MG132, ALLN, or lactacystin) or autophagy inhibitor (bafilomycin A1 or chloroquine) 3 h before harvest. Total cell lysates were subjected to IP analysis with an anti-Myc antibody, and the precipitates were separated in 6% SDS-PAGE followed by immunoblotting analysis with the indicated antibodies. All data show representative images of three independent experiments.

**Figure 3 ijms-22-07572-f003:**
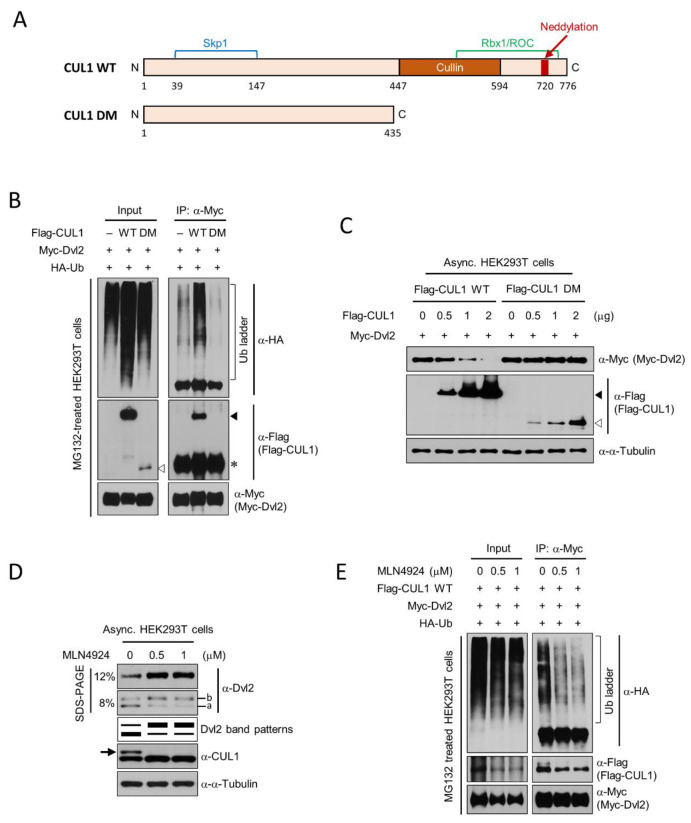
Neddylation-dependent CUL1 activation induces Dvl2 ubiquitination. (**A**) Schematic diagram of the domain structure of CUL1 WT and CUL1 DM (an inactive truncation mutant of CUL1) used in this study. The amino acid numbers, potential interacting proteins, and domain names are indicated. (**B**) Asynchronously growing HEK293T cells transfected with empty vector (-), *pFlag-CUL1 WT* (WT), or *pFlag-CUL1 DM* (DM) were cotransfected with *pMyc-Dvl2* and *pHA-Ub*. The resulting cell lysates were subjected to IP with an anti-Myc antibody. The precipitates were separated in 6% SDS-PAGE followed by immunoblotting analysis with the indicated antibodies. The filled or empty arrowhead indicates the Flag-CUL1 WT or DM, respectively. Asterisk, mouse IgG heavy chain. (**C**) Different amounts of *pFlag-CUL1 WT* or *DM* plasmids were cotransfected with *pMyc-Dvl2* into asynchronously growing (Asnyc.) HEK293T cells. The resulting cell lysates were subjected to immunoblotting analysis with the indicated antibodies. The filled or empty arrowhead indicates Flag-CUL1 WT or DM expression, respectively. A constant amount of total DNA was transfected into the cells. (**D**) Different amounts of neddylation inhibitor (MLN4924) were treated with asynchronously growing HEK293T cells for 6 h. The resulting cell lysates were subjected to immunoblotting analysis with the indicated antibodies. Immunoblotting with the anti-α-tubulin antibody represents the loading control of proteins. The arrow indicates neddylated CUL1. Fast- and slow-migrating Dvl2 forms are referred to as the “a” and “b” forms, respectively. (**E**) Asynchronously growing HEK293T cells transfected with *pFlag-CUL1*, *pMyc-Dvl2*, and *pHA-Ub* for 24 h were treated with different amounts of neddylation inhibitor (MLN4924) for 6 h; 0 μM MLN4924 represents control DMSO treatment. To prevent proteasomal degradation, the resulting cells were treated with MG132 3 h before harvest. Total cell lysates were subjected to an IP assay with an anti-Myc antibody. The precipitated beads were subjected to 6% SDS-PAGE followed by immunoblot analysis with the indicated antibodies. All data show representative images of three independent experiments.

**Figure 4 ijms-22-07572-f004:**
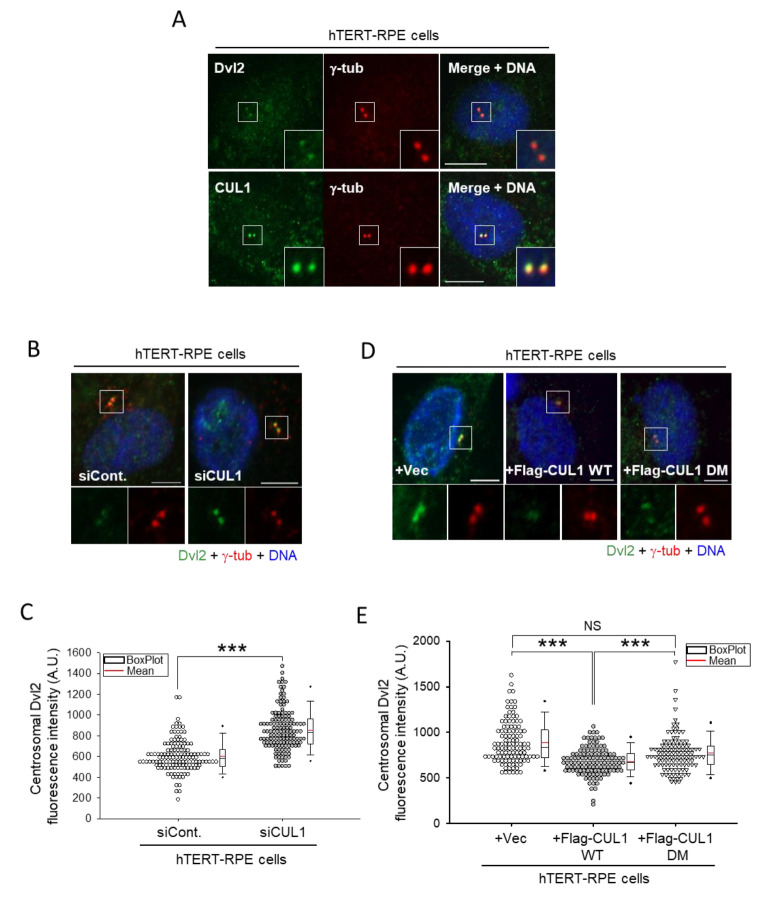
Amount of centrosomal Dvl2 was regulated by CUL1 activity. (**A**) Asynchronously growing hTERT-RPE cells were double-immunostained with anti-Dvl2 and anti-γ-tubulin antibodies or anti-CUL1 and anti-γ-tubulin antibodies. DNA was stained with 4′,6-diamidino-2-phenylindole (DAPI). Scale bar, 10 μm. (**B**,**C**) hTERT-RPE cells transfected with either control (siCont.) or *CUL1* siRNA (siCUL1) were serum-starved for 48 h. The resulting cells were subjected to immunostaining with anti-Dvl2 and anti-γ-tubulin antibodies (**B**). Scale bar, 10 μm. DNA was stained with DAPI. The signal intensities of centrosomal Dvl2 were measured using ZEN 2 blue edition (Carl Zeiss) software and shown in a dot-density graph with a box plot (**C**). *** *p* < 0.001 (unpaired two-tailed *t*-test). More than 100 cells were measured to determine the signal intensity of centrosomal Dvl2. (**D**,**E**) hTERT-RPE cells transfected with control vector (-), *pFlag-CUL1 WT*, or *DM* were serum-starved for 24 h. The resulting cells were subjected to immunostaining with anti-Dvl2 and anti-γ-tubulin antibodies (**D**). Scale bar, 10 μm. DNA was stained with DAPI. The signal intensities of centrosomal Dvl2 were measured using ZEN 2 blue edition (Carl Zeiss) software and shown in a dot-density graph with a box plot (**E**). *** *p* < 0.001; NS, not statistically significant (one-way ANOVA). More than 100 cells were measured to determine the signal intensity of centrosomal Dvl2. All data show representative images of three independent experiments.

**Figure 5 ijms-22-07572-f005:**
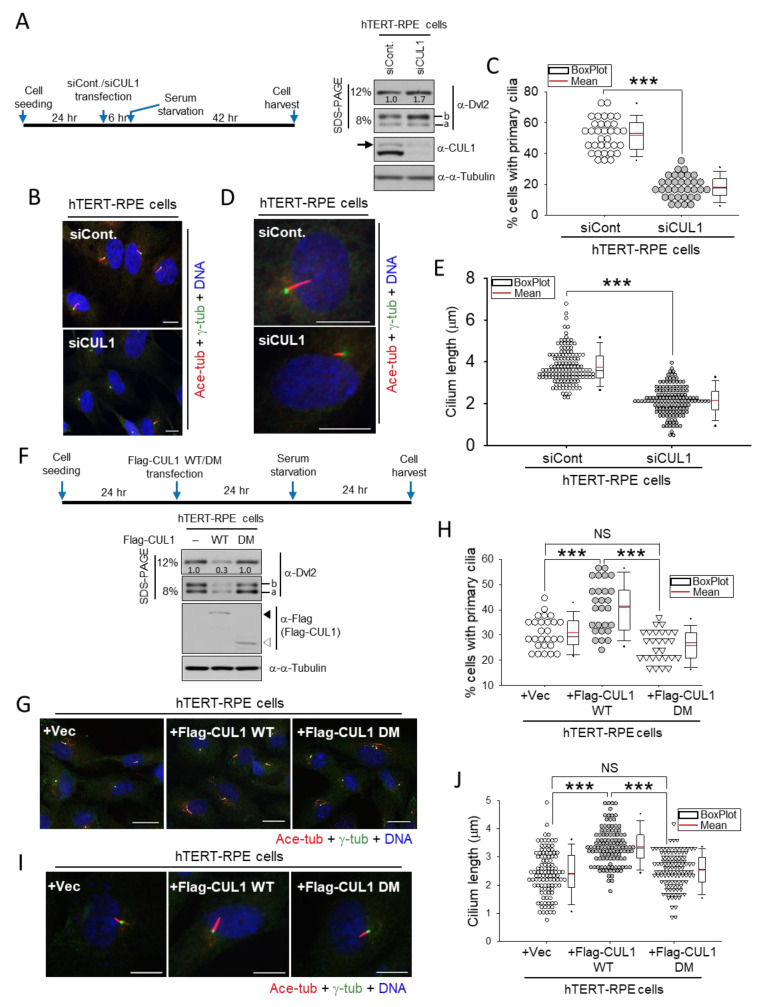
CUL1 promotes primary ciliogenesis. (**A**) hTERT-RPE cells transfected with either control (siCont.) or *CUL1* siRNA (siCUL1) were serum-starved for 48 h according to the experimental schedule (**left**). The resulting cell lysates were subjected to immunoblotting analysis with the indicated antibodies (**right**). Immunoblotting with the anti-α-tubulin antibody represents the loading control of proteins. The band intensities of endogenous Dvl2 were measured using ImageJ and normalized to α-tubulin, and the relative values are marked below the bands. The arrow indicates neddylated CUL1. Fast- and slow-migrating Dvl2 forms are referred to as the “a” and “b” forms, respectively. (**B**–**E**) The same conditioned cells as in (**A**) were subjected to immunostaining with anti-acetylated α-tubulin and anti-γ-tubulin antibodies (**B**,**D**). Scale bar, 10 μm. DNA was stained with DAPI. Cells containing primary cilia were counted (**B**,**C**), and the length of each primary cilium was measured (**D,E**). Each symbol in the dot-density graph represents a percentage (%) of ~40 cells (**C**). More than 100 cells were measured to determine the length of cilia (**E**). *** *p* < 0.001 (unpaired two-tailed *t* test). (**F**) hTERT-RPE cells transfected with control vector (-), *pFlag-CUL1 WT* (WT), or *DM* (DM) were serum-starved for 24 h according to the experimental schedule (**top**). The resulting cell lysates were subjected to immunoblotting analysis with the indicated antibodies (**bottom**). Immunoblotting with the anti-α-tubulin antibody represents the loading control of proteins. The band intensities of endogenous Dvl2 were measured using ImageJ and normalized to α-tubulin, and the relative values are marked below the bands. Fast- and slow-migrating Dvl2 forms are referred to as the “a” and “b” forms, respectively. The filled or empty arrowhead indicates the Flag-CUL1 WT or DM, respectively. (**G**–**J**) The same conditioned cells as in (**F**) were subjected to immunostaining with anti-acetylated α-tubulin and anti-γ-tubulin antibodies (**G**,**I**). Scale bar, 10 μm. DNA was stained with DAPI. Cells containing primary cilia were counted (**G**,**H**), and the length of each primary cilium was measured (**I**,**J**). Each symbol in the dot-density graph represents a percentage (%) of ~40 cells (**H**). More than 100 cells were measured to determine the length of cilia (**J**). *** *p* < 0.001; NS, not statistically significant (one-way ANOVA). All data show representative images of three independent experiments.

**Figure 6 ijms-22-07572-f006:**
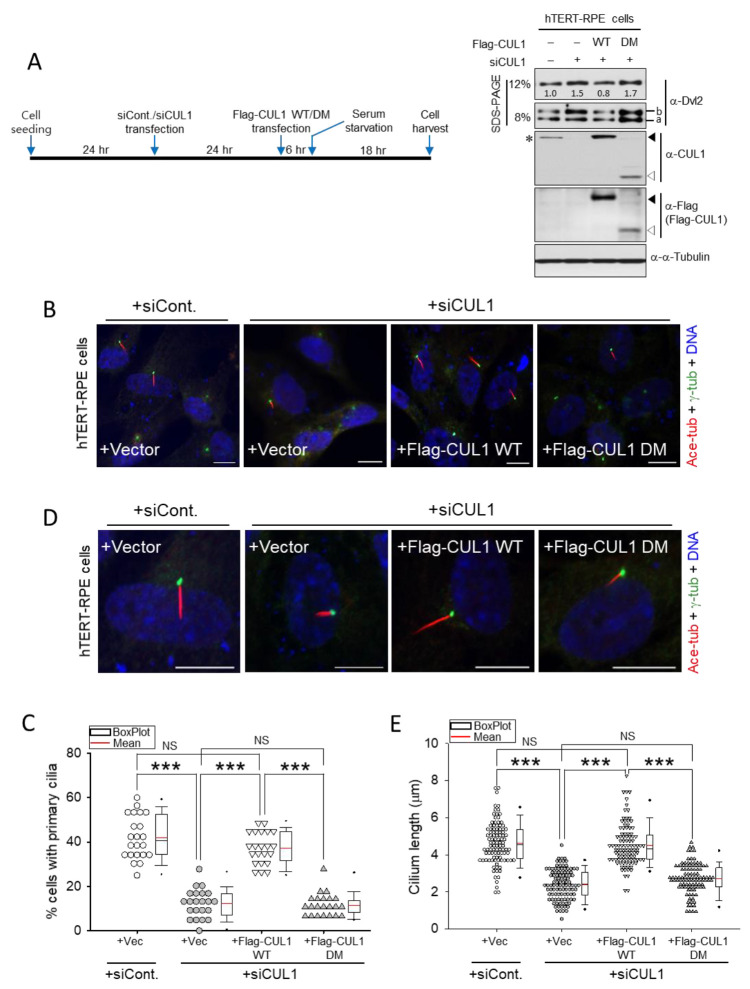
Restoration of primary ciliogenesis by CUL1 expression in CUL1-depleted hTERT-RPE cells. (**A**) hTERT-RPE cells transfected with either control or *CUL1* siRNA were transfected with control vector (-), *pFlag-CUL1 WT* (WT) or *DM* (DM). Cells were then serum-starved for 24 h according to the experimental schedule (**left**). The resulting cell lysates were subjected to immunoblotting analysis with the indicated antibodies (**right**). Immunoblotting with the anti-α-tubulin antibody represents the loading control of proteins. The band intensities of endogenous Dvl2 were measured using ImageJ and normalized to α-tubulin, and the relative values are marked below the bands. Fast- and slow-migrating Dvl2 forms are referred to as the “a” and “b” forms, respectively. Asterisk indicates endogenous CUL1. The filled or empty arrowhead indicates the Flag-CUL1 WT or DM, respectively. (**B**–**E**) The same conditioned cells as in (**A**) were subjected to immunostaining with anti-acetylated α-tubulin and anti-γ-tubulin antibodies (**B**,**D**). Scale bar, 10 μm. DNA was stained with DAPI. Cells containing primary cilia were counted (**B**,**C**), and the length of each primary cilium was measured (**D**,**E**). Each symbol in the dot-density graph represents a percentage (%) of ~30 cells (**C**). More than 100 cells were measured to determine the length of cilia (**E**). *** *p* < 0.001; NS, not statistically significant (one-way ANOVA). All data show representative images of three independent experiments.

## Data Availability

Not applicable.

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
