# Peer review of "Cullin 1 (CUL1) Promotes Primary Ciliogenesis through the Induction of Ubiquitin-Proteasome-Dependent Dvl2 Degradation"

_ijms, 2021, doi:10.3390/ijms22147572_

Round 1

Reviewer 1 Report

Using knockdown and overexpression-based assays, Kim et al identified the involvement of Cul1 in the degradation of Dvl2, a key regulator of primary ciliogenesis. Overall the manuscript is well-organized and easy to read. I have the following suggestions:

Major:

  1. Is there any possibility to identify the corresponding F-box adaptor proteins for Cul1-based Dvl2 degradation. Did you test Fbxl13, which is also hit in the M/S? I believe it would be a necessary attempt whatever the results come out.
  2. To preliminarily explore the relationship between Cul1 and Cul3-dependent Dvl degradation, I believe it is worthwhile to investigate whether Cul1 affects the expression of components involved in Cul3-dependent Dvl degradation.
  3. It is well established that neddylation generally enhances the activity of cullin ubiquitin ligases via inducing conformational changes on cullin CTD and RING proteins. Considering that Cul3 was also reported to regulate Dvl2 degradation, it is not a good idea to claim the role of neddylation in Cul1-specific Dvl2 ubiquitination. In addition, Cul1 K720 point mutant would be a better control than Cul2 DM mutant.
  4. The authors observed specific effect of Cul1 on phosphorylated Dvl2, which is interesting because SCF ligases often recognize the phosphodegron for degradation. I suggest the authors make efforts to discuss it more.

Minor:

  1. To better understand the story, background on SCF regarding basic components and regulators should be added to the introduction. Several recent reviews might be helpful to organize (PMID: 31898220; PMID: 28271482).
  2. Line 102. “a significant amount” is not suitable for a qualitative pulldown assay especially when more sensitive WB was used.
  3. Line 229. I feel there is only a slight difference between 1 and 0.5 μ

Reviewer 2 Report

IJMS/Kim et al: Cullin 1 (CUL1) promotes primary Ciliogenesis through the induction of ubiquitin-proteasome-dependent Dvl2 degradation

General comments: This is a thorough and elegant study that establishes a new regulatory pathway related to cilliogenesis. The conclusions are based on well designed and executed experiments that verify each finding by multiple approaches and from multiple angles. The presentation and interpretation of data is clear. Figures and figure legends are well prepared and easy to understand. Experimental procedures are well explained. I recommend this study for publication as proposed, with only a few minor suggestions.

  • Figure 2C: Consider to add information about the total DNA used in each experiment. Increasing amounts of Flag-CUL1 WT DNA were used in transfections, with ‘0’ ug indicating control DNA. Was the total DNA amount constant in each set and equal to the highest Flag-CUL1 DNA amount used? It would be of interest to provide this information, as only then the transfection efficiencies would be similar regardless of what specific amount of Flag-CUL1 DNA amount was used. This suggestion applies to all the experiments performed with varying/increasing DNA amounts.
  • Figure 3B: Consider to point the empty arrowhead toward the CUL1 mutant position in the input (where the signal is undisputable) rather than in the IP (where it overlaps with IgG and appears to be dislocated by the huge IgG excess in proximity).
  • Consider different interpretation of the ‘unexpected’ Dvl2a and Dvl2b degradation in cells exposed to starvation (page 10, lanes 297-298: “It is probably due to the excessive response resulting from the overexpression system” (and page 14, lanes ; and 417-420, a similar statement). Starvation is known to activate autophagy and to temporarily increase proteasomal involvement in cellular proteostasis (to recycle the essential amino acids from proteins). In my mind, a more likely explanation is thus that starvation affects the relevant proteolytic pathways, also affecting proteolysis of the two Dvl2 forms.  
  • Consider adding a comment on whether the siRNAs sequences used to downregulate CUL1 would be expected to have no effect on CUL3. This could be included in the relevant part of discussion (page 13, lanes 384-395) and/or Materials and Methods (page 14, lanes 441-443).
  • Re: proteasome inhibitors. The proteasome inhibition was convincing and executed properly, with short treatment times compensating for the relatively high (but broadly used and accepted) inhibitors' concentrations. However, my suggestion for future experiments would be to use carfilzomib (CFZ), the only proteasome inhibitor currently known to have no off target effects.
